# Isorhodopsin: An Undervalued Visual Pigment Analog

**Willem J. de Grip [1,2,\*] and Johan Lugtenburg [1]**

[1]   Leiden Institute of Chemistry, Department of Biophysical Organic Chemistry, Leiden University,
     2300 RA Leiden, The Netherlands; lugtenbu@chem.leidenuniv.nl
[2]   Radboud Institute for Molecular Life Sciences, Radboud University Medical Center,
     6500 HB Nijmegen, The Netherlands
\*    Correspondence: w.j.de.grip@lic.leidenuniv.nl

**Abstract:** Rhodopsin, the first visual pigment identified in the animal retina, was shown to be a photosensitive membrane protein containing covalently bound retinal in the 11-*cis* configuration, as a chromophore. Upon photoexcitation the chromophore isomerizes in femtoseconds to all-*trans*, which drives the protein into the active state. Soon thereafter, another geometric isomer—9-*cis* retinal—was also shown to stably incorporate into the binding pocket, generating a slightly blue-shifted photosensitive protein. This pigment, coined isorhodopsin, was less photosensitive, but could also reach the active state. However, 9-*cis* retinal was not detected as a chromophore in any of the many animal visual pigments studied, and isorhodopsin was passed over as an exotic and little-relevant rhodopsin analog. Consequently, few in-depth studies of its photochemistry and activation mechanism have been performed. In this review, we aim to illustrate that it is unfortunate that isorhodopsin has received little attention in the visual research and literature. Elementary differences in photoexcitation of rhodopsin and isorhodopsin have already been reported. Further in-depth studies of the photochemical properties and pathways of isorhodopsin would be quite enlightening for the initial steps in vision, as well as being beneficial for biotechnological applications of retinal proteins.

**Keywords:** 9-*cis* retinal; photochemistry; membrane protein; visual pigments; vibrational spectroscopy; mechanism; photoisomerization; analogs; quantum yield

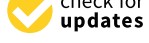



## 1. Introduction

Rhodopsin is the first visual pigment identified, namely in the rod photoreceptor cells of the vertebrate retina [1,2]. It was shown to be a photosensitive membrane protein containing a vitamin A derivative (retinal) in the 11-*cis* configuration, as a chromophore, covalently bound to an internal lysine side chain via a protonated Schiff base (Figure 1) [2–6]. Upon absorption of a visual photon, the chromophore isomerizes in femtoseconds into the all-*trans* configuration, which triggers a sequel of conformational changes in the protein, driving it within milliseconds into the active state, which triggers the photoreceptor cell towards a neural response [5,7–9]. This response—membrane hyperpolarization—is processed by the bipolar and ganglion cells in the retina, and transmitted via the optic nerve towards the visual cortex, where it is translated within several tens of milliseconds into a visual response [10,11].

(a) 11-*cis* chromophore

(b) all-*trans* retinal

(c) 9-*cis* chromophore

(d) all-*trans* retinoxime

**Figure 1. Chemical structures of the chromophore configurations in rhodopsin and isorhodopsin, and of free retinals:** The retinals are covalently bound to a lysine residue as a protonated retinylidene Schiff base with the 11-*cis*,15-*anti* configuration in rhodopsin (**a**) and the 9-*cis*,15-*anti* configuration in isorhodopsin (**c**) in the "dark state" (in photophysical terminology, "ground state"). The chromophore is photoexcited into the all-*trans* configuration, which terminally relaxes and is eventually released as all-*trans* retinal (**b**). In the presence of hydroxylamine (HO–NH$_2$) retinal is converted into retinoxime (**d**). In rhodopsin, as well as in isorhodopsin, the chromophore contains a 6-s-cis ring-polyene chain orientation and the 12-s-trans conformation. Because of the extended conjugated chain (alternating single (C–C) and double (C=C) bonds), the excitation energy is reduced, and retinal and retinoxime absorb in the UV-A region (300–420 nm). The protonated Schiff base with a lysine residue, generated upon covalent binding of the retinal with opsin (**a,c**), leads to further reduction in the excitation energy, red-shifting the absorbance maxima to 486 nm (isorhodopsin) and 498 nm (rhodopsin). Next to the lysine residue, the opsin binding pocket contains a docking site for the ring element and for the C19 methyl group. Binding into the pocket forces a twist and torsion in the conjugated chain. The main twist in the rhodopsin chromophore resides in the C10–C13 segment, with a torsion angle of circa 40°. The main twist in the isorhodopsin chromophore is divided over the C6–C9 and the C9–C11 segments, with a torsion angle of circa 20° over the latter. For further details and references, refer to the text.

The all-*trans* chromophore is eventually released as all-*trans* retinal by hydrolysis of the Schiff base, leaving the apoprotein opsin [7,12]. During this process, the opsin relaxes into a conformation close to that of rhodopsin, and has very little signaling activity left [13–16]. To regenerate the rhodopsin state, the all-*trans* retinal has to be isomerized into the 11-*cis* configuration which, when transported to the opsin site, will spontaneously recombine with opsin, generating rhodopsin, both in vivo and in vitro [1,5,6,17]. In the vertebrate retina, several systems for the production of 11-*cis* retinal are operational in the retina pigment epithelium (RPE) and cone photoreceptor cells, employing both enzymatically driven and photoactivated isomerases [18,19].

It was soon realized that the rod photoreceptor pigments had evolved for very-high-sensitivity (single-photon) black-and-white (scotopic) vision [20]. In the vertebrate realm, the absorbance maxima of the rod pigments range from 440 to 520 nm, depending on their habitat [21,22]. For color discrimination, a closely related set of pigments had evolved that cover the entire UV–deep-red region (350–640 nm), and are confined in the vertebrate cone photoreceptors [23–25]. In addition, it has become obvious that the visual pigments in the invertebrate realm are structured upon the same basic principle, but are slightly differently composed [26,27]. Specifically, they do not release the photogenerated all-*trans* chromophore, but require a second light pulse to recover the original 11-*cis* chromophore (bistable pigments) [26,28]. Thanks to the phenomenal developments in the current genome era (e.g., genome mining, recombinant DNA technology, heterologous expression), the

animal rhodopsin family has expanded spectacularly over the last few decades, and the majority of these new species are bi-stable pigments involved in light-driven physiology in a wide array of tissues [27,29–32]. As far as we know now, all animal rhodopsins are involved in signaling processes, using GTP-binding proteins as second messengers, and constitute an important subfamily in the superfamily of G-protein-coupled receptors (GPCRs) [33,34]. Simultaneously, another rapidly expanding rhodopsin-like clan was discovered in unicellular organisms (the microbial rhodopsins) [35–37]. These pigments instead exploit an all-*trans* > 13-*cis* photoisomerization with thermal relaxation to the ground state, and mainly function as light-driven ion pumps or channels, but can also have a sensory or enzyme-modulating activity [38–40]. In view of their significantly different composition, photochemistry, and functionality, the microbial pigments are now classified as type-1 rhodopsins, and the animal pigments as type-2 rhodopsins.

Soon after the initial characterization of the animal rod pigment rhodopsin, it was observed that another geometric isomer—9-*cis* retinal—could also be stably incorporated into the binding pocket of rod opsin, generating a photosensitive protein with a slightly blue-shifted absorbance maximum (486 nm versus 498 nm for the native bovine pigment) [6,17]. This analog was termed isorhodopsin. Isorhodopsin could trigger the same photoactivation cascade in the protein, but with much lower photosensitivity. Since in that time segment 9-*cis* retinal had not been detected as a chromophore in any of the many animal visual pigments studied, isorhodopsin was passed over as an exotic and little-relevant rhodopsin analog. Consequently, few in-depth studies of its photochemistry and activation mechanism have been carried out. More recently, however, it was demonstrated that in certain pathologies of the retina isorhodopsin is generated in vivo, and that administration of a relatively stable 9-*cis* retinal derivative can restore vision in mammals that have a genetic defect in 11-*cis* retinal biosynthesis [41–43].

In this review, we aim to illustrate that it is unfortunate that isorhodopsin has received little attention in visual research and literature. So far, elementary differences between the photoexcitation of rhodopsin and isorhodopsin have already been reported. Apart from its evident medical relevance, further in-depth studies of the photochemical properties and pathways of isorhodopsin and its analog pigments, using a variety of biophysical techniques and in silico calculations, would be quite enlightening for the initial steps in vision, as well as being beneficial for biotechnological applications of retinal proteins.

## 2. Isorhodopsin—General Aspects

While the designation "isorhodopsin" in principle can be used to classify any retinal protein with 9-*cis* retinal as its chromophore, we here use it exclusively in its original context, i.e., the combination of 9-*cis* retinal with the bovine rod opsin. In other cases, we use the parent pigment with an "iso-" prefix.

In solution, 11-*cis* retinal experiences steric hindrance between the C18 methyl group and H8, and between the C20 methyl group and H10 (cf. Figure 1). As a consequence, its conformation is in equilibrium between torsional conformers around the 6–7, 10–11, and 12–13 single bonds [44]. In the binding site of the apoprotein (opsin), 11-*cis* retinal adopts a tight-fitting 6-s-cis,12-s-trans,15-*anti* conformation [45–55]. Nevertheless, 11-*cis* retinal spontaneously and rapidly incorporates into bovine opsin, generating rhodopsin [6,17]. The binding pocket contains specific anchoring space in docking sites for the ring element and the C19 methyl group, in the covalent Schiff base connection to the lysine ε-amino group, and to a lesser extent for the C20 methyl group [37,56–68]. In order to fit optimally, the 11-*cis* chromophore adopts a twist in the 10–11=12–13 segment (cf. Figure 1), while all single bonds are not significantly perturbed as compared to free 11-*cis* retinal [45,48,69–79]. The Schiff base linking the chromophore to the opsin is protonated (cf. Figure 1), and this positively charged element is stabilized by a complex counterion, which consists of a negatively charged counterion (Glu113) in combination with a hydrogen-bonded network including nearby opsin residues and bound water molecules [62,70,80–92]. The shift in the absorbance maximum, from circa 440 nm for a free protonated Schiff base

of retinal to 498 nm in rhodopsin, strongly depends on the structure of the H-bonded network and counterion complex involving variable electrostatic interactions with the protonated Schiff base, and to a lesser extent on the properties of protein residues in the opsin binding pocket [62,81,93–97].

Compared to the 11-*cis* isomer, free 9-*cis* retinal is less troubled by internal steric hindrance [6,98]. Nevertheless, the rate of rhodopsin synthesis from opsin and 11-*cis* retinal is significantly higher than that of isorhodopsin synthesis from opsin and 9-*cis* retinal (5–10-fold, depending on conditions [6,17,99]). This is probably related to the circa 5 kcal/mol higher energy content of isorhodopsin relative to rhodopsin [99,100]. The ring–chain connection of the 9-*cis* chromophore is also 6-s-cis [101], but the induced fit of 9-*cis* retinal in the opsin binding pocket is suboptimal. The 9-*cis* chromophore is faced with strain in the ring and the C19-methyl docking sites, in the 10–11 and 14–15 single bonds, and in the 7=8 and 9=10 double bonds [48,56,57,62,69,78,102–105]. The C20 methyl group is also slightly displaced as compared to the rhodopsin chromophore [57]. In fact, the chromophore constellation in isorhodopsin has been referred to as an "induced misfit" [102]. The torsion in the 9=10 double bond of the 9-*cis* chromophore (circa 20°) is smaller than the torsion in the 11=12 bond of the 11-*cis* chromophore (circa 40°). On the other hand, the torsion in the 9=10 double bond of the 11-*cis* chromophore as well as in the 11=12 double bond of the 9-*cis* chromophore is relatively small (<5° and circa 10°, respectively) [57,79]. A very important difference between the two chromophores is that the largest torsion in the 9-*cis* chromophore resides in the 7=8 double bond, while in the 11-*cis* chromophore it resides in the 11=12 double bond.

Very informative in this context is vibrational spectroscopy (FTIR and resonance Raman). Of special relevance are the fingerprint region (1200–1300 cm$^{-1}$), with a specific signature for each geometric isomer, and the hydrogen-out-of-plan (HOOP) region (800–1000 cm$^{-1}$). In particular, when two HOOP vibrations are coupled in a *trans*-ethylenic bond, they degenerate and split up into an out-of-phase (Bg) and an in-phase (Au) component. The *trans*-coupled Au HOOP ($^{H}C=C_{H}$) and the *cis*-coupled A2 HOOP ($^{H}C=C^{H}$) absorb in the 900–990 cm$^{-1}$ region [72,106–109], and their amplitude depends on the torsion in the ethylenic bond [71,72,108,110,111]. In the Raman scattering spectra, these coupled HOOPs turn up with their intensity depending on the torsion in the ethylenic bond and the loss of local symmetry. In particular, the Au HOOP in a flat structure has very low intensity in Raman spectra, but is strongly enhanced upon torsional deflection of the ethylenic bond. In infrared absorbance spectra, the coupled HOOPs are always active, but again their intensity strongly depends on the torsion in the ethylenic bond. Assignment of vibrational bands to structural elements is assisted by selective $^{13}$C or $^{2}$H labeling of the chromophore via organic synthesis, and can subsequently also validate physical computation [71,78,106,108,112–114]. Furthermore, solid-state NMR spectroscopy can also provide detailed structural information on the chromophore and its interaction with the binding pocket environment, using properly labeled and/or otherwise modified retinals [55,69,74,76,89,102,113–122].

While the absorbance maxima of free 11-*cis* and 9-*cis* retinal and their Schiff bases are very close [6,17], the maximum of isorhodopsin (486 ± 2 nm) is significantly blue-shifted from that of rhodopsin (498 ± 2 nm). This probably relates to the somewhat different arrangement of the complex counterion and the interaction with the binding pocket residues [102,104]. So far, a blue shift has been observed in all visual iso-pigments tested [101,123], except for the primate blue visual pigment, which apparently cannot stably bind 9-*cis* retinal [124,125].

## 3. Analog Pigments

Early on, it was realized that investigating the effects of modified retinals on pigment properties would enable significant information to be unraveled. For instance, data such as incorporation rates, spectral properties, photochemistry, G-protein activation, etc., could provide intimate insight into aspects such as binding pocket restraints, spectral tuning, and

photoactivation mechanisms [45,58,63,72,112,123,126–132]. The majority of these analogs are based on the 11-*cis* isomer, but analogs of other isomers have also been studied—predominantly the 9-*cis* isomer (e.g., [45,101,105,113,123,126,127,131,133,134]).

It is essential to verify that the isomeric state of the analog (for instance, 9-*cis* or 11-*cis*) is retained upon incubation with opsin, since the incorporation rate is often strongly reduced, and analog pigment formation may take many hours instead of minutes (e.g., [131,135–137]). This is commonly performed by quantitative extraction of the chromophore as the oxime derivative (Figure 1), followed by liquid chromatographic identification [138,139]. For instance, while free 9,11-di*cis* and 9,13-di*cis* retinal are stable in organic solvents, upon incubation with opsin thermal isomerization occurs, and both isorhodopsin and the di*cis*-pigment are generated [131,135,136]. Furthermore, the 9,13-pigment slowly isomerizes thermally into isorhodopsin, and the 9,11-pigment is decomposed upon incubation with hydroxylamine [131,140]. For identification of these iso-pigments, several analytical data are available: the significant differences in the spectral and vibrational profiles of the 9-*cis*, 9,11-di*cis*, and 9,13-di*cis* retinals [141]; the different chromatographic profiles of the corresponding oxime derivatives; and the distinctly different absorbance maxima of the corresponding pigments (486, 472 and 478 nm, respectively [131,136]).

Hydroxylamine ($HO–NH_2$) reacts with free retinals (R–C=O) under the formation of the oxime derivative (Figure 1), which is much less susceptible to thermal isomerization [17,139,142]. Hydroxylamine can also react with Schiff bases generating the oxime, but most rod pigments are stable or decompose only very slowly in the presence of hydroxylamine. This demonstrates that in the rod pigments the binding pocket is very poorly accessible to this reagent. To be able to extract the chromophore as the stable oxime, it is then essential to denature the protein chemically in the presence of a large excess of hydroxylamine [139]. On the other hand, in most cone pigments and many analog pigments the Schiff base is more accessible to hydroxylamine, and the chromophore is eventually released as the oxime derivative [17,24,123]. When this is the case for analog pigments of rhodopsin and isorhodopsin, it indicates a poor fit of the analog in the binding pocket, rendering the Schiff base region accessible to small hydrophilic reagents.

An alternative way to identify the isomeric state of the chromophore, allowing in situ observation, is to employ vibrational spectroscopy. The fingerprint region of the vibrational spectra, reflecting single C–C bond stretching vibrations, is very characteristic of the isomeric configuration of retinals and retinylidene chromophores. This is usually sufficient for identification [78,106,141,143]. In this review, we focus on the HOOP vibrations ($^HC=C_H$) and ($^HC=C^H$), since they provide information on the structure of the chromophore, as explained above. In rhodopsin and isorhodopsin, this pertains to the 7=8 and 11=12 HOOPs (Figure 1). In the 7,8-dihydro analogs, the 7=8 double bonds are reduced to a single bond, and only the 11=12 HOOP remains. Table 1 shows the vibrational assignment for these HOOPs in rhodopsin, isorhodopsin, and selected analog pigments, based on resonance Raman and FTIR spectroscopy. When a vibration cannot be observed or is weak, it indicates that the corresponding ethylenic bond is largely free from or low in torsion. Here, we only present rhodopsin and isorhodopsin, but the 11-*cis* and 9-*cis* pigments of the red cone pigment iodopsin and of squid and octopus rhodopsin exhibit HOOP vibrational frequencies that are very close to the rod pigments [47,144–148]. The data in Table 1 show several interesting features. In agreement with other studies, the 11-*cis* chromophore in the native rhodopsin only shows strain in the 11=12 double bond. In contrast, the 9-*cis* chromophore in isorhodopsin exhibits less strain in the 11=12 double bond, but clear indication of strain in the 7=8 double bond. This probably reflects the poor fit of the ring and the C19 methyl group in their docking site [57,102,104]. In the 7,8-dihydro pigments, the 7-8 bond is now single and less restrained, which increases the flexibility around the ring–chain connection. This does not release the strain in the 11=12 double bond, and therefore does not make much difference for the 11-*cis* chromophore, but the 7,8-dihydro 9-*cis* chromophore probably fits better than the unmodified 9-*cis* chromophore (see Section 4). As expected, elimination of the C20 methyl group, resulting in the 13-desmethyl pigment, reduces the

strain in the 11=12 double bond and, thus, the intensity of the 11=12 coupled HOOP in the 11-*cis* chromophore. In contrast, the addition of a methyl group at C10, resulting in the 10-methyl pigment, perturbs the fit of the 11-*cis* chromophore and increases the strain in the 11=12 double bond.

**Table 1.** HOOP frequencies of selected 11-*cis* and 9-*cis* pigments.

| Retinal | | 11-*cis* Pigments | | 9-*cis* Pigments | | References |
|---|---|---|---|---|---|---|
| Modification | Chemical Structure * | HOOP Frequency (cm$^{-1}$) # | | HOOP Frequency (cm$^{-1}$) # | | |
| | | 7=8 | 11=12 | 7=8 | 11=12 | |
| - |  | u | 969 | 959 | 969 (w) | [48,71,78,105,108,144,149–152] |
| 7,8-Dihydro |  | - | 971 | - | 967 (w) | [56,105,134] |
| 13-Desmethyl |  | u | 975 (w) | n.a. | n.a. | [64,153] |
| 10-Methyl |  | u | 954 (s) | n.a. | n.a. | [130] |
| 9-Cyclopropyl |  | 977 (w) | 988 (w) | u | 974 | [105,154] |
| 3,4-Dehydro (retinal A2) |  | n.a. | n.a. | 960 (s) | 970 (w) | [155] |

* Chemical structures shown are those of the all-*trans* isomers. # u: undetected, w: weak, s: extra strong, n.a.: not available, -: not present.

Particularly remarkable is the strong and opposite effect on the intensity of the HOOP vibrations upon replacing the C19 methyl group with the more voluminous cyclopropyl group (Table 1). In the case of the 11-*cis* chromophore, this clearly enhances the 7=8 HOOP, reflecting more strain in the 7=8 double bond, and weakens the 11=12 HOOP, reflecting less strain in the 11=12 double bond. This reflects a much poorer fit in the binding pocket, as compared to the unmodified 11-*cis* retinal. In contrast, in the case of the 9-*cis* chromophore, the strain in the 7=8 double bond is strongly reduced, while the strain in the 11=12 bond is increased. This more closely resembles the HOOP profile of the unmodified 11-*cis* chromophore, and suggests that the 9-*cis* cyclopropyl analog creates a better induced fit in the binding pocket than its parent 9-*cis* retinal. This is discussed further in Section 4.

In this context, as well as in view of its photochemical properties, it is also noteworthy to look at the 3,4-dehydro analog (Table 1). In fact, the 11-*cis* isomer of this analog is also found in vertebrate visual pigments—in particular, those of freshwater and coastal vertebrates [123,156]. The 3,4-dehydro-retinal is also denoted as retinal A2, to differentiate it from the abundant retinal (Figure 1)—also denoted as retinal A1. The more extended conjugated chain in retinal A2 further decreases the excitation energy, thereby red-shifting the absorbance band by about 20 nm. As a chromophore, retinal A2 red-shifts the absorbance maximum by 20–40 nm in rod pigments and up to 70 nm in cone pigments, as compared to retinal A1 [17,22,101,157]. The extra double bond in the ring (Table 1) also induces more stiffness in the ring dynamics. The HOOP frequencies of the 11-*cis* 3,4-dehydro-chromophore have not been reported but, as discussed below, it probably fits equally well as the 11-*cis* A1 chromophore. However, the intensity of the 7=8 HOOP vibration in the

9-*cis* 3,4-dehydro-chromophore is enhanced relative to the 9-*cis* A1 chromophore (Table 1), suggesting an increase in the strain and in the "misfit" in the binding pocket.

## 4. Photochemistry

The photochemistry of iso-pigments has mainly been studied in isorhodopsin. Among the animal rhodopsins, the photochemical cascade of bovine rod rhodopsin has been investigated in most detail, with a variety of techniques (ultrarapid optical and Raman spectroscopy of the kinetics of selected transitions, crystallography and vibrational and NMR spectroscopy of stabilized photo-intermediates, advanced in silico computational procedures, etc.) (e.g., [79,82,102,158–166] and references therein). A global scheme is presented in Figure 2, and we first discuss this in some detail. The photocascade is triggered by isomerization of the photoexcited chromophore towards the all-*trans* configuration, which results in formation of the first photo-intermediate bathorhodopsin (Batho) within femtoseconds. This intermediate is red-shifted from the ground-state rhodopsin [167,168]. The efficiency of this isomerization reaction is astonishing, occurring within femtoseconds with a quantum yield of 0.65 (i.e., two out of three photons are productive) [169–172]. Rapid kinetic studies indicate that in free 11-*cis* retinal, as well as its protonated Schiff base, photoisomerization is more selective towards the all-*trans* isomer, and occurs within picoseconds, with quantum yields between 0.2 and 0.3 [173–175]. Hence, in the rhodopsin context, this conversion is surprisingly efficient and selective in 11-*cis* to all-*trans*. This is facilitated by the constraints of the binding site and the twist in the C10-C13 segment of the chromophore.

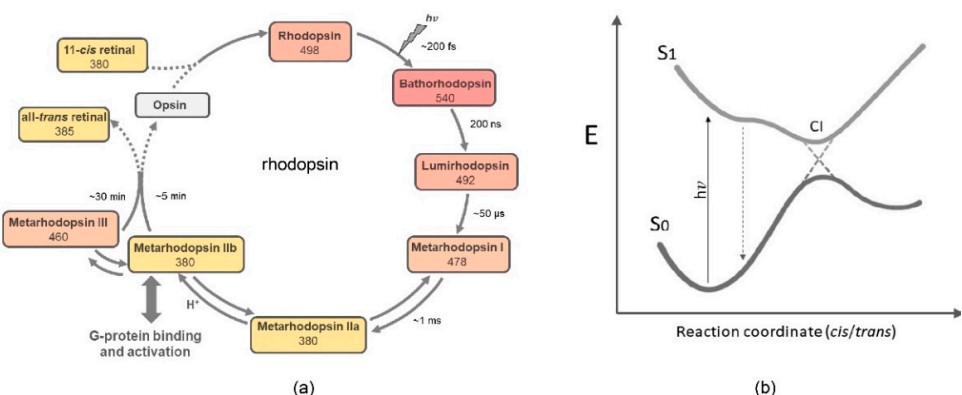

**Figure 2. Global presentation of the predominant photochemical pathway in rhodopsin:** (**a**) The 11-*cis*,15-*anti* chromophore configuration in rhodopsin is photoisomerized into all-*trans*. It remains all-*trans* in all photo-intermediates, except for Meta III, which contains a 15-*syn* protonated Schiff base [58]. The early photo-intermediates also contain a protonated Schiff base, and relax thermally to deprotonated Meta II states. In many visual pigments, Batho is in equilibrium with a blue-shifted intermediate (BSI) [176,177]. (**b**) Simplified schematic of a conical intersection where the excited chromophore at the S1 potential energy surface can cross over to the S0 potential energy surface of the photoproduct. The S1 surface can contain thermal transitions, as suggested for isorhodopsin (see text). For rhodopsin, very early interstate mixing between the S1 and S2 states has been proposed, which would lead to excited state population splitting [178].

A very effective combination of a selectively deuterated chromophore at the 11 and/or 12 position with femtosecond spectroscopy and advanced quantum chemical computation resolved many issues in the photoisomerization process of bovine rhodopsin [179]. The global picture arises that, after photoexcitation of the chromophore into the Franck–Condon state, it rapidly relaxes along a barrierless trajectory on the potential surface, to a minimal-energy conical intersection (Figure 2). Here, productive resonance of the electronic wave packet at the excited-state potential surface, with torsional and HOOP vibrational modes in the twisted C10–C13 segment of the 11-*cis* chromophore, can prime very effective crossover to a ground-state energy surface, generating a hot all-*trans*oid state (photorhodopsin) after

tens of fs [171,178,180]. This relaxes thermally within about 200 fs into the photoproduct Batho, which contains a still highly twisted all-*trans* chromophore, but is stable below 130 K [78,108,167]. Recent advanced QM/MM simulations suggest excited-state population splitting resulting in several waves of productive crossover to the photoproduct's ground state in the first 150 fs, promoted by vibronic coupling, vibrational phase matching, and electrostatic interactions in the binding site [178].

At room temperature, the circa 35 kcal of excitation energy stored in Batho [181–183] drives further relaxation via several intermediates until the metarhodopsin IIa-IIb equilibrium is reached within ms. This relaxation process subtly rearranges chromophores, protein residues, and H-bonded hydrated networks up to the metarhodopsin stage [12,184–188], where the Schiff base transfers its proton, the counterion and another Glu residue at the intracellular side of the protein become protonated, and an interhelical activity switch reshuffles helical segments to open up binding residues for the G protein [12,189–195]. The chromophore is subsequently slowly released via hydrolysis of the Schiff base to generate the nearly inactive apoprotein opsin [7,13,15,196]. In vivo, however, the active state is rapidly inactivated through phosphorylation by rhodopsin kinase, followed by arrestin binding, which blocks further activation of the G protein [197]. As documented in the Introduction, in order to retrieve the rhodopsin pigment, the opsin end-state of the photocascade has to be refurbished with 11-*cis* retinal, so as to spontaneously regenerate rhodopsin in accord with the circa 10 kcal lower energy level of the holoprotein [100].

Soon after its discovery, it was already established that, as from bathorhodopsin, excitation of isorhodopsin yields the same photocascade as rhodopsin, albeit with significantly lower photosensitivity (i.e., quantum yield of 0.25 [99,154,172,198]). As a result, isorhodopsin also signals the G protein, albeit with reduced kinetics at low light intensity [43,199]. Overall, the isorhodopsin photocascade significantly differs in several aspects from that of rhodopsin, namely, the pigment (re)generation rate, the photoisomerization mechanism, the kinetics and quantum yield, and the temperature dependence. Pigment regeneration is discussed in Section 2. The other aspects are probably interrelated, and are discussed here in context.

It was established that the chromophore in Batho is still bound via a protonated Schiff base, but has an all-*trans* configuration and a highly twisted conformation in the single bonds, whereby the C19 and C20 methyl groups and the ring are barely displaced [57,82,167,184,200,201]. Very informative in this context are the vibrational spectra—in particular, the fingerprint region and the HOOP and torsional region [78,107,108,153,202]. The general consensus is that upon photoexcitation rhodopsin and isorhodopsin generate the same first photo-intermediate—bathorhodopsin (Figure 2). Indeed "both" Bathos exhibit very similar spectral and vibrational properties [48,70,72,85,107,108,144,146,150,152,153,167,176,203–205]. However, it has been reported that the photo-intermediates Batho and Lumi may subtly differ between rhodopsin and isorhodopsin with respect to their absorbance maxima, molar absorbance, vibrational features, and the position of the C20 methyl group [57,104,205–207].

The exceptionally high quantum yield of rhodopsin—which is hardly affected by wavelength or temperature [169,170,208]—in combination with its ultrafast photoisomerization [179,180,202,209–213], is interpreted as reflecting a vibronically coherent photochemical process. This would involve productive vibronic coupling of nuclear vibrational modes of the isomerizing 10-11=12-13 segment and the excited electronic wave packet, resulting in a barrierless S1 potential energy surface trajectory to a minimal-energy conical intersection, with very efficient crossover to the S0 surface of the *trans* photoproduct via a bicycle pedal motion [79,153,166,170,178–180,207,210,214–220]. The important contribution of the coupled 11=12 HOOP vibrational phase is perfectly illustrated by analysis of deuterated analogs, where the 11,12-dideutero derivative ($^D$C=C$^D$) even slightly increases the quantum yield (Table 2), while the mono-deuterated analogs substantially decrease the quantum yield [179], similar to other 11- or 12-mono-substituted analogs [149,221]. Deuteration does not affect the rate of Batho formation (Table 2). The twist in the 10-11=12-13 segment of the polyene chain in the ground state would sub-

stantially prime the entire process. However, the chromophore is tightly constrained in the binding pocket, especially with respect to the ring and C19 methyl group, and at the femtosecond-to-picosecond timescale other motions in the binding pocket are restricted to the hydrated hydrogen-bonded network and very small displacements of the protein residue side chains [222–224]. Hence, 11-*cis* ⇒ *trans* isomerization would require a kind of concerted motion in the 10–14 segment, for which volume-conserving "bicycle-pedal", "hula-twist", and "wring-a-wet-towel" trajectories have been proposed, and slight displacement of the C20 methyl group is required [79,111,225]. A bicycle pedal motion now seems to be generally accepted [178,207,217].

However, for isorhodopsin, this situation is markedly different. First of all, the photoisomerization rate is reduced significantly, since Batho formation only levels off at about 600 fs, compared to the 200 fs for rhodopsin [158,209,213,226]. In addition, the quantum yield of isorhodopsin is not only significantly reduced (Table 2), but is also strongly temperature- and wavelength-dependent. At 77 K, the quantum yield is reduced to 0.16 at wavelengths ≤ 460 nm, and decreases further to 0.09 at 540 nm [99,169]. This has been interpreted as a small activation barrier along the S1 potential energy surface. However, it should be realized that compared to the 11-*cis* chromophore, the strain in the 9-*cis* chromophore is distributed quite differently. Most of the strain in the 9-*cis* chromophore is located in the 7=8 *trans* double bond, of which the skeletal vibrational modes and A2 HOOP are counterproductive for isomerization of the 9=10 *cis* bond—also because of the fixed position of the C19 methyl group. Furthermore, the strain in the 9=10-11=12-13 segment is relatively modest (cf. Section 2). Hence, we also take the view, that the photoisomerization trajectory of isorhodopsin must deviate from that of rhodopsin [57,79,207]. In this context, it is relevant to inspect Table 2.

First of all, Table 2 presents some exceptions to the common feature that 9-*cis* pigments are blue-shifted relative to the corresponding 11-*cis* pigments. For instance, when the 5=6 or 7=8 double bonds are reduced to single bonds, the corresponding 9-*cis* and 11-*cis* pigments have very similar spectral properties ([227] and Table 2), suggesting that the increase in flexibility around the ring segment improves the induced fit of the 9-*cis* chromophore. Nevertheless, it has been reported that the Batho ↔ BSI equilibrium behaves differently in the 9-*cis* 5,6-dihydropigment as compared to the corresponding 11-*cis* pigment [227]. More spectacular is the effect of a cyclopropyl substituent at C9. Here, the 11-*cis* pigment is blue-shifted, while the 9-*cis* pigment is significantly red-shifted, and even passes by the 11-*cis* pigment (Table 2).

**Table 2.** Optical and photochemical parameters of selected 11-*cis* and 9-*cis* pigments [#].

| Retinal * | Isomer | λmax Pigment (nm ± 0.02) | Photochemistry | | References |
|---|---|---|---|---|---|
| | | | Time for Batho to Max | Quantum Yield | |
| Native (A1) | 11-*cis* | 498 | 200 fs | 0.65 ± 0.02 | [72,99,158,169,170,172,208,209,213,226] |
| | 9-*cis* | 486 | 600 fs | 0.25 ± 0.04 | |
| 7,8-Dihydro A1 | 11-*cis* | 426 | - | 0.68 ± 0.06 | [105,127,134,228,229] |
| | 9-*cis* | 428 | - | 0.39 ± 0.04 | |
| 9-Cyclopropyl A1 | 11-*cis* | 492 | - | 0.08 ± 0.04 | [105,154] |
| | 9-*cis* | 504 | - | 0.39 ± 0.04 | |
| 10-Methyl A1 | 11-*cis* | 506 | - | 0.55 ± 0.07 | [130,198,230] |
| | 9-*cis* | 498 | - | <0.2 | |
| 11,12-D2 A1 | 11-*cis* | 498 | 200 fs | 0.69 ± 0.02 | [179] |
| | 9-*cis* | - | - | - | |
| 12-D A1 | 11-*cis* | 498 | 200 fs | 0.48 ± 0.03 | [179] |
| | 9-*cis* | - | - | - | |
| 13-Desmethyl A1 | 11-*cis* | 496 | 400 fs | 0.46 ± 0.04 | [72,137,221,231–234] |
| | 9-*cis* | 486 | - | - | |
| 14-Fluoro A1 | 11-*cis* | 528 | - | 0.55 ± 0.10 | [149,198,235,236] |
| | 9-*cis* | 510 | - | 0.40 ± 0.06 | |
| 3,4-Dehydro A1 (A2) | 11-*cis* | 518 | - | 0.63 ± 0.04 | [155,208,237,238] |
| | 9-*cis* | 500 | - | 0.10 ± 0.02 | |

[#] All data taken at room temperature. -: "data not available". * For full chemical structures, see Table 1.

The effects on the quantum yield of the photoisomerization are also instructive. The substantial decrease in the quantum yield of the mono-deuterated 11-*cis* pigment (12-D) is similar to that of the 11-D analog [179], as explained above. Where tested, the quantum yield of the 11-*cis* pigments is higher in most cases than that of the corresponding 9-*cis* pigments. A significant increase in the quantum yield of analog 11-*cis* pigments has not yet been observed, but for 9-*cis* pigments this is clearly the case for the 7,8-dihydro, 9-cyclopropyl, and 14-F analogs. Again, the 9-cyclopropyl pigments are exceptional, as the 11-*cis* quantum yield drops precipitously while the 9-*cis* quantum yield increases substantially (Table 2).

How can all of these data fit into a plausible photoisomerization trajectory for isorhodopsin? The astonishing decrease in the quantum yield of the 11-*cis* 9-cyclopropyl-pigment evokes the deduction that an increase in the twist in the 7=8 double bond with a decrease in the twist in the 11=12 double bond (cf. Table 1) is counterproductive for photoisomerization. While this thesis does need further experimental and computational underpinning, it is supported by the 7,8-dihydro data, showing that with a flexible 7–8 single bond presenting solitary C–H vibrations the quantum yield of the iso-pigment is substantially enhanced. It is further supported by our observation that in 9-cyclopropyl-isorhodopsin a decrease in the 7=8 strain with an increase in the 11=12 strain (cf. Table 1) is accompanied by an increase in the quantum yield (Table 2). Unfortunately, we do not have the HOOP profiles of the 14-F analogs, but further studies on these pigments are also strongly recommended.

We can therefore conclude that in isorhodopsin the 11=12 *trans* double bond is intimately involved in the photoisomerization of the 9=10 *cis* double bond. In fact, QM/MM trajectory simulations of excited isorhodopsin in combination with ultrarapid spectroscopy predicted up to 23% production of a 9,11-di*cis* photoproduct [79,207]. The more recent article [207] derives two distinct trajectories on the S1 potential energy surface, leading to two distinct conical intersections: a faster unproductive trajectory, producing isorhodopsin and 9,11-di*cis* rhodopsin, and—due to sterical interaction with the protein—a slower trajectory producing the all-*trans* photoproduct via a forward pedal rotation. However, 9,11-di*cis* rhodopsin has never been observed as a photoproduct of isorhodopsin. Moreover, we would expect stronger sterical interaction with the protein in the case of the 9-cyclopropyl derivative, thereby reducing the quantum yield. In the 7,8-dihydro- and 9-cyclopropyl-isorhodopsin, an absent or reduced twist in the 7=8 *trans* double bond, in combination with a larger twist in the 11=12 *trans* double bond, results in an increase in the quantum yield. Hence, we can infer that the vibrational phase of the 11=12 *trans* double bond plays an important role in the photoisomerization of the 9-*cis* chromophore in isorhodopsin. Our thesis is that on the S1 potential surface of excited isorhodopsin, a (major?) trajectory leads to a conical intersection and crossover under *trans* ⇒ *cis* isomerization of the 11=12 double bond, resulting in a hot ground-state 9,11-di*cis*-like chromophore. In the overheated binding pocket with strong vibrational activity, this can thermally return to the original state (isorhodopsin), or can transition via a bicycle pedal mechanism to a highly twisted all-*trans*oid Batho-like photoproduct. This would slow down the entire process, and may also explain the increase in stimulated emission [178,207]. It is also consistent with the very slow incorporation of 9,11-di*cis* retinal into the opsin at room temperature, the higher energy level of the 9,11-di*cis* pigment, and its susceptibility to attack by hydroxylamine [79,131,140], indicative of a very unstable induced fit in the 9,11-di*cis* pigment. It is also conceivable that cryotemperatures and lower-energy photons, which reduce the vibrational excitement, promote return to the isorhodopsin ground state rather than isomerization to all-*trans*, resulting in a reduction in the isomerization quantum yield. In conclusion, we propose that in the excited binding pocket of isorhodopsin a hot 9,11-di*cis*oid photoproduct is generated, and a thermal trajectory leads to significant all-*trans* formation. Both trajectories are enhanced by the torsion and vibrational activity of the 11=12 bond.

In this context, it is noteworthy that the enhanced quantum yields of the isorhodopsin analogs all level out around 0.4 (Table 2)—still substantially below that of rhodopsin. This may be related to the intrinsically slower torsional velocity and less straightforward dynamics in the 9-*cis* isomer [79,207,226,239].

Finally, we note again that 3,4-dehydroretinal shows interesting behavior (Tables 1 and 2). The quantum yield of 3,4-dehydrorhodopsin is hardly reduced. Although HOOP frequencies are not available, this is consistent with the concept that the isomerization trajectory of rhodopsin is concentrated in a fast and productive bicycle pedal motion in the C9-C13 reaction coordinate, and is scarcely influenced by the dynamics of the ring [57,178,207,210]. In contrast, the quantum yield of 3,4-dehydro-isorhodopsin is significantly reduced relative to isorhodopsin itself (Table 2). Again, this is consistent with the stronger 7=8 HOOP intensity since, as discussed above, the twist in the 7=8 double bond is counterproductive in the iso-pigments. This would further impede the photoisomerization trajectory in 3,4-dehydro-isorhodopsin towards the all-*trans* photoproduct via the 9,11-di*cis*oid photoproduct, reducing the quantum yield. In further support, similar to isorhodopsin, the quantum yield of 3,4-dehydro-isorhodopsin is temperature-dependent, dropping to 0.05 at 80 K [155].

## 5. Physiological and Medical Relevance

Iso-pigments not only can be generated with animal opsins, but also have been observed in the microbial opsin family, that without exception employs all-*trans* retinal for its chromophore, which is photoisomerized to the 13-*cis* configuration, and eventually thermally reverts to the all-*trans* ground state [36,38]. For instance, the microbial prototype bacteriorhodopsin (BR) generates a 9-*cis* photoproduct upon sustained illumination, and this phenomenon may also occur in other microbial rhodopsins [36,240,241]. The putative microbial proton pump MR (middle rhodopsin) can even slowly incorporate 9-*cis* retinal, generating the iso-pigment [242]. Nevertheless, most microbial opsins do not generate iso-pigments, in contrast to most animal opsins.

Initially, visual pigments were isolated and laboriously purified from natural sources [17,22,123,243,244]. For solubilization, mild natural surfactants such as bile acids and digitonin were used [1], since cone pigments in particular were not very stable when dissolved with the commercially available surface-active agents [17,244,245]. Later on, mild synthetic detergents such as CHAPSO, OG, and DDM were developed [246–248], and now dominate the retinal protein field. This was also of great help when the recombinant DNA era arrived in the 1980s and proficient heterologous hosts for the production of the apoprotein (opsin) in cell culture were developed [249–253]. In order to generate the holoproteins, 11-*cis* retinal had to be supplied to the cell culture, but this compound has several disadvantages, including low thermal stability in biological environments [17,142], not being commercially available, and elaborate purification after chemical synthesis or after photoproduction from all-*trans* retinal [17,254,255]. Conversely, 9-*cis* retinal has better thermal stability [6] and is commercially available; hence, it seems to be a more suitable alternative.

This would not cause any problem for bi-stable rhodopsins, such as invertebrate visual pigments and melanopsins, since after photoexcitation they do not release all-*trans* retinal, but halt in a stable metarhodopsin state that can be photoisomerized to the corresponding 11-*cis* pigment [26,27,256]. In fact, the recent first crystal structure of an arthropod rhodopsin (jumping spider) was achieved with the iso-pigment [257]. When investigating the slower pathways in a rhodopsin photocycle, iso-pigments would also be a proper alternative, except for mutants or analog pigments, where the photocascades may deviate [113,133,154]. When using iso-pigments in vitro or in cell culture, one must take into account the much slower incorporation rate of 9-*cis* retinal into the opsin, the lower thermodynamic stability, and the lower photosensitivity of iso-pigments [258–262]. In particular, when using 9-*cis* retinal in physiological studies—for instance, when scanning for chaperones stabilizing rhodopsin mutants, investigating rhodopsin oligomerization in vivo, or studying organoid differentiation—, it is essential to verify essential data with 11-*cis* retinal [263–269].

Unexpectedly, 9-*cis* retinal has come to play an important role in treating certain pathologies of the vertebrate retina. Genetic defects in enzymes involved in the pathways responsible for furnishing the rod photoreceptor with 11-*cis* retinal—such as the isomerohydrolase RPE65, 11-*cis* retinol dehydrogenases, or lecithin retinol acyltransferase (LRAT)—can lead to night

blindness, retinitis pigmentosa, or retinal degeneration [270–272]. Subcutaneous injection of 11-*cis* retinal in mutant mice did not properly relieve the symptoms [273]. When it was observed that mutant mice slowly produced isorhodopsin when kept in darkness, which enhanced the rod photoreceptor response [43,199], the option presented itself to test application of the more stable alternative 9-*cis* retinal. Indeed, early intervention in mutant mice with intravenously or intraperitoneally administered 9-*cis* retinal attenuated retinal loss of function and degeneration in a dose-dependent manner [274–276]. For oral delivery, 9-*cis* retinal was not very suitable, but 9-*cis* retinyl acetate—the acetate ester of the corresponding alcohol—proved to be sufficiently stable, and was rapidly absorbed and intracellularly hydrolyzed to 9-*cis* retinol, which did reach the retina [42]. Following intracellular enzymatic oxidation to 9-*cis* retinal, isorhodopsin was generated, which in mutant mice again attenuated retinal pathology [42,277,278]. An elaborate study in human patients also showed beneficial effects of oral delivery of 9-*cis* retinyl acetate, yielding sustained improvement in visual field and visual acuity of up to 70% of the patients [41,279]. Further improvement in pharmacological treatment was achieved by using a slow-release chitosan-9-*cis*-retinal conjugate [280]. This prodrug was tested in mice and dogs, and shown to be slowly absorbed from the gastrointestinal tract, resulting in sustained plasma levels of 9-*cis*-retinol and dose-dependent recovery of visual function lasting for several weeks [280].

Other physiological effects of retinols, such as production and function of the corresponding retinoic acids, are beyond the scope of this paper, and we refer to a relevant recent review [281].

## 6. Overview and Prospects

We expect to have provided compelling evidence that 9-*cis* retinal, while photophysiologically much less abundant and photophysicochemically much less exposed than 11-*cis* retinal, presents quite an attractive profile for detailed further studies.

For medical applications, the use of 9-*cis* retinal or derivatives will probably be restricted to retinal defects in 11-*cis* retinal production, since its potential in other retinal pathologies seems to be limited [282,283]. Testing analogs for this purpose is a complex option, since the physiological stability and pigment regeneration rate of the analog, along with the spectral properties and photoisomerization quantum yield of the corresponding iso-pigments, should not be problematic. For instance, 9-*cis* 7,8-dihydroretinal would improve the quantum yield, but would significantly blue-shift the photosensitivity of the rod photoreceptor, potentially disturbing the visual sensation. On the other hand, a locked analog, preventing photoisomerization, has been shown to have a protective effect against light-induced retinal degeneration [284], and to selectively block rod opsin consumption of chromophores while largely sparing cone opsins, prolonging cone-dependent vision [280].

For studies at a physiological, cellular, or in vitro level, 9-*cis* 7,8-dihydroretinal would actually be an interesting alternative to 9-*cis* retinal itself, since the spectral shift would then not be very disturbing, and the induced fit and the photoisomerization quantum yield would be closer to those with 11-*cis* retinal. In fact, it should be worthwhile to test whether the combination of the 7,8-dihydro element with a 14-F substitution would redshift the corresponding iso-pigment closer to rhodopsin without reducing the quantum yield (Table 2).

However, most interesting are the photochemical mechanism and potential of the iso-pigment family. We have presented that, while rhodopsin is almost evolutionarily "finalized"—exhibiting ultrafast, nearly unidirectional, and very efficient photo-activation—isorhodopsin is much less effective. Nevertheless, the fascinating feature of isorhodopsin is that its mechanism and performance can be adapted and/or improved by retinal analogs. From that point of view, isorhodopsins offer great promise, which definitely needs to be further explored.

We have postulated an adapted photoisomerization trajectory for isorhodopsin, which should be verified by similar experimental and theoretical approaches as used for rhodopsin. Femtosecond spectroscopy, supported by advanced QM/MM computation, in combination

with di- and mono-deuteration of the 7=8 double bond and/or of the 11=12 double bond, should yield principal information on their contribution to the rate and quantum yield of Batho formation by isorhodopsin. We postulate that mono-deuteration of the 7=8 double bond would reduce its counterproductive effect, thereby enhancing the quantum yield, while mono-deuteration of the 11=12 double bond would slow down generation of the 9,11-di*cis* product, thereby probably reducing the quantum yield and the rate of Batho formation. Similar studies with 3,4-dehydro, 7,8-dihydro-, and 9-cyclopropyl-isorhodopsin could reveal whether the same mechanistic principle is indeed active in these pigments, which should also make their photoproduct quantum yield temperature-dependent. In this context, analysis of 8–18, 11–13, or 12–14 ring-fused isorhodopsin analogs would also be very informative [101,285]. For instance, in the 8,18 methano-analog, the 7=8 double bond is mono-substituted and nearly flat, and we would postulate an increase in the quantum yield for the corresponding iso-pigment. The photoisomerization trajectory we propose for isorhodopsin would be perturbed or prohibited in the 11–13 and 12–14 ring-fused isorhodopsin analogs, and it would be very relevant to investigate whether alternative photoisomerization trajectories can be selected.

From another perspective, it would be very valuable if femtosecond structural studies became available for animal (iso)rhodopsins, as has recently been accomplished for microbial rhodopsins using femtosecond XFEL crystallography [222,286]. For animal pigments, it would probably be advisable to use cryo-EM in combination with the more easily prepared and more stable nanodisc environment [287], in an ultrarapid mode when possible. Eventually, in silico ab initio structural model building exploiting the newly available algorithms [288–290] will allow rapid testing of the chromophoric potential of theoretical promising retinal analogs for (iso)rhodopsins [291]. Atomic insight into the photoisomerization trajectories of isorhodopsins will be of utmost importance, not only from a photochemical point of view, but also for the design of retinal proteins with selected properties—both for bioengineering purposes and in optogenetics [292,293].

**Author Contributions:** Conceptualization: J.L. and W.J.d.G.; writing—original draft preparation, W.J.d.G.; writing—review and editing, W.J.d.G. and J.L. All authors have read and agreed to the published version of the manuscript.

**Funding:** This research received no external funding.

**Institutional Review Board Statement:** Not applicable.

**Informed Consent Statement:** Not applicable.

**Data Availability Statement:** Not applicable.

**Acknowledgments:** The authors express their gratitude to Srividya Ganapathy (University of California at San Diego, School of Medicine) for her help with preparation of the figures.

**Conflicts of Interest:** The authors declare no conflict of interest.

## Abbreviations

| | |
|---|---|
| BR | Bacteriorhodopsin |
| CHAPSO | 3-[(3-Cholamidopropyl)dimethylammonio]-2-hydroxy-1-propanesulfonate |
| Cryo-EM | Cryo-electron microscopy |
| DDM | Dodecylmaltoside |
| DFT | Density functional theory |
| FTIR | Fourier-transform infrared |
| Glu | Glutamate residue in the protein |
| HOOP | Hydrogen out-of-plane vibration |
| LRAT | Lecithin retinol acyltransferase |
| NMR | Nuclear magnetic resonance |
| OG | Octyl glucoside |
| QM/MM | Quantum-mechanical/molecular-mechanical |
| XFEL | X-ray free-electron laser |

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
