# Peer review of "Isorhodopsin: An Undervalued Visual Pigment Analog"

_2079-6447, doi:10.3390/colorants1030016_

Round 1
Reviewer 1 Report
Please see the attached dcomments.

Author Response
We appreciate the careful evaluation by the reviewer. Our response to the reviewer comments is attached. Please note, that the line numbering in the revised manuscript is perturbed because of the text amendments. Deleted or relocated text is presented by "Track Changes"in the balloons in the margin.

Reviewer 2 Report
de Gripe's paper is a detailed review comparing the functional properties of the most studied natural chromophore of vertebrate and invertebrate visual pigments, 11-cis retinal, and the nearly-natural chromophore 9-cis retinal, as well as several other chromophores. The comparative aspect of the paper allows a better understanding of the subtle molecular mechanisms of chromophore isomerization.
Author Response
We appreciate the evaluation by the reviewer. Since the reviewer did not post any comments or questions, we cannot include any response. Please note, that the line numbering in the revised manuscript is perturbed because of the text amendments. Deleted or relocated text is presented by "Track Changes"in the balloons in the margin.
Reviewer 3 Report
The genesis of this paper is an extensive and well-detailed review about one of the limited studied rhodopsin analogue: isorhodopsin. The authors reported, through a parallel comparison with rhodoppsin, the differences in photoexcitation, and provided a literature review about their photochemical properties and pathways.
Minor points:
We suggest the incorporation of a summarizing figure in section "2. Isorhodopsin - General Aspects". We surmise that there are many details described in the main text about single-double bonds, dihedral angles, HOOP, torsions, HBN that can confuse to the reader. We believe that an extra figure comparing the described parameters about 11-cis and 9-cis chromophores will be very helpful to improve the quality of the manuscript.
Section "3. Analog pigments". Even if the English language and style during all the manuscript was satisfactory, in this specific section we found some very long sentences that me the reading comprehension a little bit difficult making us read the section several times for its correct interpretation. We would appreciate if this section can be reevaluated/rewritten.
In this same section, we would like to suggest two modifications:
1) Related with "Table 1. HOOP frequencies of selected 11-cis and 9-cis pigments" we believe that question mark symbol (?) for not available HOOP frequencies may confuse the reader. Please, consider another symbol.
2) From line 196 to 204, retinal A1 and retinal A2 are named. An inexperienced reader can easily not recognize the difference between these two retinal. We suggest to highly/explain with a few extra sentences the differences between this two retinals and why they are important. Please, also consider to introduce the following reference:
Luk, H. L.; Bhattacharyya, N.; Montisci, F.; Morrow, J. M.; Melaccio, F.; Wada, A.; ... ; Olivucci, M., Modulation of thermal noise and spectral sensitivity in Lake Baikal cotton fish rhodopsins. Scientific Reports 2016, 6 (1), 1-9.
Since "Figure 2. Global presentation presentation of the predominant photochemical pathway in rhodopsin" described in section "4. Photochemistry" is not an original figure created by the authors, we recommend the addition of "Figure adapted from... [references]" in the caption of the figure. Also, we recommend to add the following reference in this caption:
Ernest, O. P.; Bartl, F. J., Active states of rhodopsin. ChemBioChem 2002, 3 (10), 968-974.
To complete the figure, we recommend the addition of the retinal chromophore configurations in all possible photo-intermediates.

Author Response

(The authors gave the same response as above.)
